# D-SCOPE: Diffusion-based Sonar Counterfactual and Prototype Explanations

## Abstract

The harsh conditions of underwater environments pose significant challenges for effective monitoring. While using cameras is possible, they are typically limited to short ranges due to underwater visibility conditions. SOund NAvigation and Ranging (SONAR) can perceive objects at greater distances, but produces low-visibility images that are hard to interpret, even for experts. When Artificial Intelligence (AI) methods are used on these SONAR images, Explainable Artificial Intelligence (XAI) methods might help the user understand the AI outputs. Traditional explainability methods, such as saliency maps or perturbation-based visualisations, often struggle to provide informative explanations when applied to low-contrast imagery. This work introduces Diffusion-based SONAR COunterfactual & Prototype Explanations (D-SCOPE), a novel post-hoc explainability framework for SONAR image classification. Our presented approach leverages classifier-guided diffusion models, trained on two publicly available Marine Debris Forward-Looking SONAR datasets, to generate two types of visual explanations: (1) counterfactual explanations that highlight minimal semantic changes to alter a model's decision, and (2) prototype-based explanations for case-based reasoning that translate representative RGB samples into the SONAR domain, serving as intuitive visual references. For counterfactual explanations, a semi-factual explanation is generated by displaying the intermediate steps leading to a change in prediction. For the prototype-based explanation, class-specific prototypes are provided. To the best of our knowledge, this is the first approach applying diffusion-based generative models for explainability in the SONAR modality. Guided diffusion models are shown to produce high-fidelity, class-conditioned counterfactuals in challenging underwater settings. In addition, the proposed cross-domain prototype generation mechanism enhances human interpretability by bridging the gap between clear and recognisable RGB representations and SONAR imagery. Our framework is validated through qualitative and quantitative experiments as well as a controlled human evaluation. The code and the pretrained models shall be released to support further research.

## 1 Introduction

SONAR sensors play a critical role in underwater scene understanding, particularly in applications such as marine habitat monitoring, Search & Recovery, and lake surveillance (Elsayed et al., 2024). However, their outputs are often difficult to interpret, making it challenging for humans to trust deep learning systems deployed in such environments. While recent advances in underwater object recognition using deep learning have shown promising results (Wen et al., 2024; Aubard et al., 2024; Li et al., 2024; Zhang et al., 2022), these models are commonly considered as black-box models. This can raise important concerns: how do we make these models more trustworthy for end-users or provide meaningful feedback to developers when they fail? The field of XAI addresses these concerns, and it is growing particularly in high-stakes domains such as healthcare (Band et al., 2023) and autonomous driving (Atakishiyev et al., 2024), where model explainability is crucial. Research has just started to explore XAI in the context of SONAR imagery (El-Mihoub et al., 2024; Natarajan & Nambiar, 2024), yet there remains a significant gap in providing visual explanations for classification and detection results.

Nowadays, generative models, such as Generative Adversarial Networks (GANs) and diffusion models, demonstrate their ability to produce realistic images that are hard to distinguish from real data (Li et al., 2025). These capabilities enable new types of visual explanations, e.g., counterfactuals (Van Looveren et al., 2021; Jeanneret et al., 2022). A Counterfactual Explanation (CE) answers a "what-if" question:" What if this feature had a different value? Would the outcome of the model change?" However, depending on the context, CE can address broader questions (Verma et al., 2020). For example, in terms of causality, CE addresses the question "which features caused this outcome?" Answering this question can be done by identifying minimal changes needed to change the models' outcome. From an actionability perspective, it addresses the question, "which features should be modified to reach a desired outcome?" as in the case of loan approval scenarios. From a contrastive perspective, it can answer the question "why is this outcome produced by the model instead of another?" by comparing factual and counterfactual instances. Beyond counterfactuals, generative models can also support Case-Based Reasoning (CBR) through Prototype-based Explanation (PE) (Li et al., 2018). A prototype serves as a representative or "typical" example of a class, helping users understand what the model considers characteristic input patterns. Synthesising such prototypes provides an intuitive, concept-level understanding that complements instance-level counterfactual reasoning.

In this work, we introduce D-SCOPE, a post-hoc explainability framework that provides two main visual explanation methods: (1) CEs that shows how an input could be modified to change a prediction, and (2) PEs that illustrates what a typical input of a predicted class looks like. For the CEs, a query image and a user-defined target class are provided. The framework then generates a modified version of the query image classified as the target, along with Semi-factual Explanation (SE), a sequence of intermediate images showing the gradual semantic changes leading to the prediction shift. A pixel-wise difference map is also produced to highlight the regions responsible for the decision change. For the PEs, we generate class-specific prototypes by translating known examples from one modality (e.g., RGB) into another (e.g., SONAR). These synthetic images serve as cross-domain visual references that help users interpret the classification results by comparing them with a typical and semantically grounded example. Unlike saliency-based techniques that only localise attention, this method offers recognisable object appearances that reflect how the model perceives each class. While D-SCOPE is validated on SONAR data, the design of the framework is fundamentally tailored for high-noise, single-channel imagery, making its explanations highly relevant to similar low Signal-to-Noise Ratios (SNR) domains such as Synthetic Aperture Radar (SAR). To the best of the authors' knowledge, this is the first work to explore visual CEs and PEs using diffusion models in the context of SONAR imagery. Our main contributions are:

1. D-SCOPE, a unified post-hoc explainability framework for black-box classifiers in underwater SONAR imagery, leveraging generative diffusion models.
2. An empirical demonstration of guided diffusion models for generating class-conditioned counterfactuals in noisy, grayscale SONAR images.
3. A prototype-based explanation strategy for CBR, where a single guided diffusion model performs cross-domain translation (e.g., RGB to SONAR) to generate class-representative examples.
4. Qualitative, quantitative, and controlled human evaluations of the proposed D-SCOPE framework.

The rest of the paper is organised as follows: Section 2 discusses related work. Section 3 reviews the diffusion model concepts and presents our proposed methodology. Section 4 covers the experimental evaluation. Finally, Section 5 concludes with a summary and future directions.

## 2 Related Work

XAI (Gunning et al., 2019; Atzmueller et al., 2024) methods are crucial for improving the transparency and trustworthiness of deep learning black-box models. Yet, XAI addresses different user types, ranging from non-experts to domain specialists (Ford & Keane, 2022) and can be either intrinsic (through interpretable model design) (Atzmueller et al., 2024) or post-hoc (after model prediction) (Nolle et al., 2023). In this paper, we focus specifically on post-hoc explainability for classification models.

## 2.1 Post-hoc Explanation Methods

Below, we outline major categories of explanation methods for our application context.

**Feature Attribution Methods**, such as feature importance or local perturbations, are both post-hoc methods used to explain classification decisions (Ribeiro et al., 2016; Lundberg & Lee, 2017). While these methods might be useful in domains where the images are naturally interpretable by human eyes, these approaches are less effective for perceptual data like SONAR images. In SONAR, pixel-level attributions can still be uninterpretable for human eyes, especially when the images are characterised by low resolution and significant acoustic shadowing.

**Prototype-based Explanations (PE)** show examples (prototypes) that are representative of the predicted class. This method bridges the human way of thinking, addressing the statement that *"this image is identified as that class, because this image (or part of it) contains something similar to a prototype image (or part of it) that I know."* By providing such examples, the model offers an intuitive explanation to AI users, helping them to understand the decision-making process (Chen et al., 2019; Nauta et al., 2021; Rymarczyk et al., 2022; Schlinge et al., 2025).

**Counterfactual Explanations (CE)** address potential "what-if" scenarios by identifying a minimal change required to alter a model's prediction. This provides model-agnostic insights without inspecting the internal structure of the model (Wachter et al., 2017). In other words, visual CEs aim to explain a classification through minimal changes in the input image to cross the decision boundary of the classifier, while remaining interpretable to the end-users (Goyal et al., 2019).

**Semi-factual Explanations (SE)** address "even-if" scenarios that change the input image, but the classifier has to yield the same result as before (Kenny & Keane, 2021). While less commonly used, they complement CEs by showing the prediction invariance.

## 2.2 Generative Models for Counterfactuals

**Generative Models** such as GANs (Goodfellow et al., 2014) and diffusion models (Sohl-Dickstein et al., 2015) have been used to produce synthetic data that resembles real data. They have been used in image generation tasks, image editing, and semantic segmentation (Croitoru et al., 2023). In particular, GANs (Goodfellow et al., 2014) learn to generate realistic samples via adversarial training, but suffer from issues such as mode collapse and instability (Chen, 2021). Diffusion models, inspired by non-equilibrium thermodynamics, represent a newer class of generative models. They work by progressively adding noise to data and then learning to reconstruct it, enabling the generation of high-fidelity images (Sohl-Dickstein et al., 2015). Diffusion models are easier to train and more stable than GANs, although they suffer from problems such as bias amplification in the training dataset (Yang et al., 2023).

**Counterfactuals produced by Generative Models** have been used to produce CE by learning a manifold of realistic variations. They can generate samples that lie between a real image and the region where the classifier would predict a different class, creating a synthetic image that is close to another class but still shares some traits with the original. Several works have used GANs to generate CE in domains such as autonomous driving (Zemni et al., 2023), medical imaging (Mertes et al., 2022), and facial attributes manipulation (Van Looveren et al., 2021). Some of these have included human-centred evaluations, showing that such CE examples can be more effective than state-of-the-art saliency-based explanation methods (Mertes et al., 2022; Zemni et al., 2023).

More recently, diffusion models have been explored for CE generation due to their improved stability and image quality. This includes both unconditional (Jeanneret et al., 2022) and conditional (Favero et al., 2025; Kim et al., 2024; Varshney et al., 2025) diffusion approaches. These works show that diffusion-based counterfactuals are high-fidelity, realistic, and semantically consistent.

## 2.3 Explainability in SONAR Imagery

XAI for SONAR-based AI models is still relatively underexplored. Most existing methods rely on visualisations, e.g., saliency maps or self-attention, applied to classification (Natarajan & Nambiar, 2024) or detection (El-Mihoub et al., 2024; Manss & El-Mihoub, 2025). These are often not reliable, especially when the images are noisy or visually hard to interpret. Perturbation-based explanation methods have been used to investigate the factors influencing the performance of fine-tuned models on Synthetic Aperture SONAR (SAS) data (Walker et al., 2021). Other similar methods, such as submodular approaches to local explanation, have also been applied to interpret deep learning models for underwater SONAR image classification (Natarajan & Nambiar, 2024). Saliency maps have been employed to explain the classification of SAS images by convolutional neural networks (CNNs). However, evaluations have shown that end-users remain unconvinced about the effectiveness of these methods (Richard et al., 2024).

SONAR images represent acoustic backscatter intensity patterns; while they may appear visually similar to optical images, they belong to a different modality compared to RGB images. Recent works have explored translating between modalities, such as from SONAR to RGB, to improve human monitoring in low-visibility environments (Wehbe et al., 2022). While these GAN-based approaches enhance the visual interpretability of the scene for a human observer, they do not provide insights into the decision-making process of a classifier. In contrast, PEs and CEs generated via diffusion models offer a new direction by providing user-centred explanations that highlight the features driving a specific classification outcome.

## 3 Post-hoc Explainability Framework

We propose D-SCOPE, a post-hoc explainability framework for interpreting the predictions of SONAR-based classifiers. The framework does not require access to the internal structure or parameters of the model. It relies on a class-conditional diffusion model and is guided by a classifier trained on images with added Gaussian noise to generate visual explanations. These include counterfactual images that show how an object would appear if it belonged to a different class, along with the intermediate images generated before the prediction changes, which we refer to as SE. In addition, the framework supports prototype-based explanations that map optical-domain examples into their SONAR equivalents. Our framework works with classifiers and detectors trained on the Marine Debris Forward-Looking SONAR dataset (Valdenegro-Toro et al., 2025).

Figure 1 illustrates the proposed framework. The respective steps of the D-SCOPE explanation pipeline are as follows:

1. Consider a SONAR image $\mathbf{X} \in \mathbb{R}^{h \times w}$, with $h, w \in \mathbb{N}$, and a classifier $C$. Then, a class prediction $y_{\text{pred}}$ is generated by $C(\mathbf{X}) = y_{\text{pred}}$.
2. The user then asks questions about the classification, such as: *"Why is this not classified as $y_{\text{target}}$?"*
3. Accordingly, the target class $y_{\text{target}}$, along with $\mathbf{X}$, is forwarded to the explanation module, i.e. the guided diffusion model.
4. After that, the user chooses the type of explanation:
   - **Counterfactual Explanation:** A class-conditional diffusion model generates a counterfactual image $\widehat{\mathbf{X}}_{\text{cf}}$ aligned with $y_{\text{target}}$.
   - **Prototype-based Explanation:** The diffusion model performs cross-domain translation (e.g., from optical to SONAR) to generate prototype images. Then, $\widehat{\mathbf{X}}_{\text{pred}}$ is a prototype for $y_{\text{pred}}$ and $\widehat{\mathbf{X}}_{\text{target}}$ is a prototype for $y_{\text{target}}$ in a familiar modality, e.g., RGB.
5. Next, the generated image (either $\widehat{\mathbf{X}}_{\text{cf}}$, $\widehat{\mathbf{X}}_{\text{pred}}$, or $\widehat{\mathbf{X}}_{\text{target}}$) is presented to the user, who may request refinement or explore additional target classes or explanation modes.

### 3.1 Diffusion Models

Diffusion models iteratively add noise to an image and then learn how to remove the noise. To achieve this, diffusion models treat the input image as a random variable such that it can be modelled by a distribution,

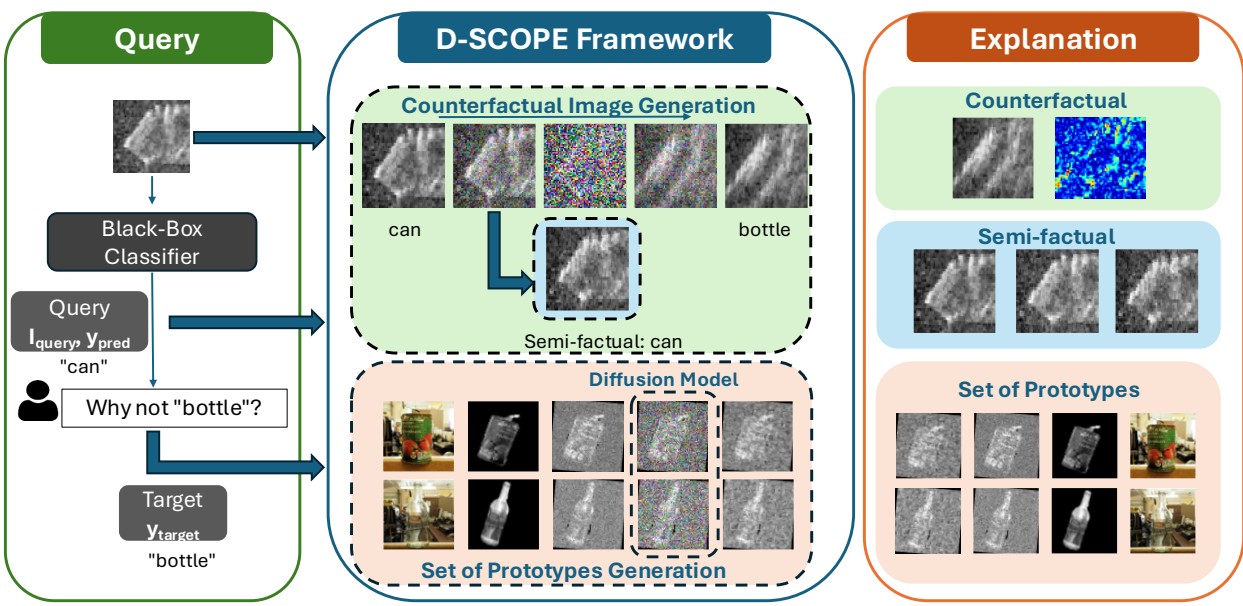

Figure 1: Overview of the explainability framework D-SCOPE for SONAR imagery.

i.e. $\mathbf{X} \sim q(\mathbf{X})$. Then, a forward Markov process is defined to add noise to the input image. The resulting noisy image can then be defined as (Nichol & Dhariwal, 2021):

$$q(\mathbf{X}_1, \ldots, \mathbf{X}_T \mid \mathbf{X}) = \prod_{t=1}^{T} q(\mathbf{X}_t \mid \mathbf{X}_{t-1}) = \mathcal{N}\left(\mathbf{X}_t; \sqrt{1 - \beta_t}\mathbf{X}_{t-1}, \beta_t \mathbf{I}\right) \tag{1}$$

where $\mathbf{X}_t$ is the noisy image at timestep $t = 1, \ldots, T$, $\beta_t \in (0, 1)$ controls the noise added at each step, and $\mathbf{I}$ is the identity matrix. Accordingly, this notation might yield $\mathbf{X}_0$, which is equal to $\mathbf{X}$.

This process is reversible (Ho et al., 2020) and is modelled by a neural network that approximates the reverse transitions

$$p_\theta(\mathbf{X}_{t-1} | \mathbf{X}_t) = \mathcal{N}(\mathbf{X}_{t-1}; \mu_\theta(\mathbf{X}_t, t), \Sigma_\theta(\mathbf{X}_t, t)), \tag{2}$$

where $\theta$ denotes the model parameters, $\mu_\theta(\mathbf{X}_t, t)$ is the predicted mean, and $\Sigma_\theta(\mathbf{X}_t, t)$ is the predicted variance at timestep $t$.

In the forward process, the image is gradually transformed into noise, such that $\mathbf{X}_T \sim \mathcal{N}(0, \mathbf{I})$. The reverse process then iteratively maps noise back to the data distribution (Nichol & Dhariwal, 2021).

To guide the generation toward a specific object class in SONAR imagery, we use a class-conditional diffusion model, where the network is conditioned on the target class $y_{\text{target}}$. Building on the works of (Dhariwal & Nichol, 2021), this enables the generation of counterfactuals and prototypes that illustrate how an object might appear if it belonged to a different class. The denoising process is learned through a neural network, which we refer to as the diffusion model $\mathcal{D}_\theta$. This network predicts the mean and variance in the reverse process $p_\theta(\mathbf{X}_{t-1} | \mathbf{X}_t)$, as in equation 2.

## 3.2 Counterfactual Explanations (CE) & Semi-Factual Explanations (SE)

When the user requests a CE for a black-box classifier's prediction, the framework generates a CE by perturbing the input image $\mathbf{X}$ with its predicted label $y_{\text{pred}}$. Noise is added to $\mathbf{X}$ to produce a noisy sample

$\mathbf{X}_t$ at timestep $t \in \{1, \dots, S\}$, where $S = 160$. This uses time respacing (Nichol & Dhariwal, 2021) to reduce the inference schedule for better computational efficiency. The noise is then removed through a reverse diffusion process guided by a class-conditional diffusion model $\mathcal{D}_\theta$. At each denoising step, the noisy classifier $\mathcal{C}_{\text{noisy}}$ provides the log-probability of the target class, $\log p_{\mathcal{C}_{\text{noisy}}}(y_{\text{target}}|\mathbf{X}_i)$, and its gradient $\nabla \mathbf{X}_i$ is used to perturb the sample toward the target class (Classifier Guidance). This produces a sequence of intermediate denoised samples $\mathbf{X}_i$, illustrating the gradual transformation from the query class toward the target class.

A candidate sample is recorded as valid if both classifiers predict the target class above a confidence threshold $\tau$. The framework selects the best counterfactual $\mathbf{X}_{\text{cf}}$ (usually corresponding to the minimal timestep that flips the prediction). It computes a pixel-wise difference map $\Delta = |\mathbf{X} - \mathbf{X}_{\text{cf}}|$ and SSIM score to quantify structural similarity to the original image.

With a fixed time respacing, the added noise to the input image is the only parameter that is changed gradually, aiming for a classifier that gives a prediction above a certain threshold. If the target class $y_{\text{target}}$ is reached, the loop breaks, and a counterfactual is generated along with a pixel-wise difference map that shows what has been changed. We compute Structural Similarity Index Measure (SSIM) scores between the query SONAR image and its counterfactual generated SONAR image to show the user how much the query image needs to be modified to produce a counterfactual image. SSIM captures low-level structural differences and provides an intuitive measure of visual shifts (see Section 4.3). The counterfactual generation method is described in Algorithm 1.

To characterise the counterfactual path in image space, the guided reverse-diffusion outputs can be viewed as a discrete trajectory $\mathcal{T} = \{\mathbf{X}_{t^*}, \mathbf{X}_{t^*-1}, \dots, \mathbf{X}_0\}$. The selected counterfactual is the endpoint of the trajectory initialised at the minimal noise injection timestep $t^*$, defined as:

$$t^* = \min \left\{ t \in \{1, \dots, S\} : \mathcal{C}(\mathbf{X}_{\text{cf}}^{(t)}) \geq \tau \ \wedge \ \mathcal{C}_{\text{noisy}}(\mathbf{X}_{\text{cf}}^{(t)}) \geq \tau \right\},$$

where $\mathbf{X}_{\text{cf}}^{(t)}$ is the candidate denoised output generated starting from noise timestep $t$. The final explanation $\mathbf{X}_{\text{cf}} = \mathbf{X}_{\text{cf}}^{(t^*)}$ at $t^*$ thus represents the valid sample generated under the minimum necessary noise perturbation. This demonstrates that the explanation is not an arbitrary reconstruction, but the final point of a non-linear, manifold-constrained trajectory that minimises displacement from the source image while entering the target decision region.

Our approach employs a class-conditional diffusion model, which directly conditions on the target class during training and sampling. In addition to classifier guidance during sampling, this allows more precise control over the generated images and produces counterfactuals that are well-suited for interpreting SONAR imagery.

### 3.3 Prototype-based Explanations (PE)

In addition to CE, D-SCOPE framework supports PE, where we utilise the Image-to-Image translation (I2I) concept to convert an object from the optical domain to the SONAR domain using a conditional diffusion model (Saharia et al., 2022). Moreover, instead of relying on latent vectors or image patches to justify class predictions (Chen et al., 2019), we generate cross-modal prototypes using our trained conditional diffusion model. Essentially, these cross-modal prototypes represent the same prototypical object, but in a different modality, e.g., point clouds and RGB images. The basic assumption is that prototypes in a well-known modality help humans to interpret the detections in another modality. For example, detections in a SONAR image stem from objects, which can be visualised as an optical RGB image, which in turn can be shown to an end user, but from a SONAR perspective. In other words, we can say this PE within D-SCOPE framework makes the human think: "This looks like an example that I know" (as in the optical domain) Chen et al. (2019).

The intended workflow of the system is as follows. The user asks why the classifier identifies an object in a SONAR image as, for example, a *valve* (see Figure 2). Then, the workflow starts from predefined RGB prototypes in the dataset and selects the corresponding object image. The object's background is removed,

---

**Algorithm 1** Counterfactual Generation via Guided Diffusion

---

**Require:** Query image $\mathbf{X}$, target class $y_{\text{target}}$, diffusion model $\mathcal{D}_\theta$, noisy classifier $\mathcal{C}_{\text{noisy}}$, classifier $\mathcal{C}$, guidance weight $\lambda$, confidence threshold $\tau$, max steps $S = 160$
1: **Binary search for minimal timestep** $t^* \in \{1, \ldots, S\}$
2: **while** binary search has not converged **do**            ▷ Search for minimal noise level
3:      Select candidate timestep $t$
4:      Add noise to $\mathbf{X}$ at timestep $t$
5:      **for** $i = t$ down to 1 **do**                   ▷ Denoising loop
6:          Predict mean $\mu_i$ using $\mathcal{D}_\theta(y_{\text{target}})$
7:          Compute gradient $\mathbf{g} \leftarrow \nabla_{\mathbf{X}_i} \log p_{\mathcal{C}_{\text{noisy}}}(y_{\text{target}} | \mathbf{X}_i)$
8:          Apply guidance: $\mu_i \leftarrow \mu_i + \lambda \sigma_i^2 \mathbf{g}$
9:          Sample $\mathbf{X}_{i-1} \sim \mathcal{N}(\mu_i, \sigma_i^2)$
10:      **end for**
11:      Candidate $\mathbf{X}_{\text{cf}}^{(t)} \leftarrow \mathbf{X}_0$
12:      **if** $\mathcal{C}(\mathbf{X}_{\text{cf}}^{(t)}) \geq \tau$ and $\mathcal{C}_{\text{noisy}}(\mathbf{X}_{\text{cf}}^{(t)}) \geq \tau$ **then**
13:          Record candidate $\mathbf{X}_{\text{cf}}^{(t)}$ as valid; adjust search for smaller $t$
14:      **else**
15:          Adjust search for larger $t$
16:      **end if**
17: **end while**
18: Select $\mathbf{X}_{\text{cf}}^{(t^*)}$ at minimal valid $t^*$ (smallest noise level satisfying the confidence threshold $\tau$)
19: Compute $\Delta = |\mathbf{X} - \mathbf{X}_{\text{cf}}|$, SSIM
20: **return** $\mathbf{X}_{\text{cf}}, \Delta$, SSIM

---

and the cropped region is converted into greyscale. Next, the workflow applies Contrast Limited Adaptive Histogram Equalization (CLAHE) (Pizer et al., 1987) with added noise to match the contrast characteristics of the target SONAR image. The added noise is zero-mean Gaussian noise ($\mathcal{N}(0, 0.05)$), to resemble the SONAR characteristics and to stabilise the following diffusion process, which is necessary to translate the prototype into a realistic acoustic appearance.

As illustrated in Algorithm 2, the process of prototype-based explanations is iterative. Across a range of noise levels $t \in [t_{\min}, t_{\max}]$ and geometric augmentations (flipping, scaling, and shifting), the image is injected with noise and passed through the diffusion model to generate a candidate pool $\mathcal{C}$ of denoised prototypes $\widehat{\mathbf{X}}_{\text{proto}}$. At these timesteps, the model has sufficient denoising capability to bridge the domain gap between the preprocessed prototype and a realistic sonar acoustic appearance while preserving underlying geometry.

To guide this process towards a representative explanation, we propose a selection score (Equation 3). This score evaluates the generated prototype against two reference targets. First, it evaluates against the target SONAR image (i.e., the original image queried by the user) to match orientation using cosine similarity while discouraging deviations in visual style. Second, it evaluates against the preprocessed input image (pre-diffusion) to guide visual appearance preservation via SSIM. The prototype that achieves the highest selection score is returned, prioritised by whether it meets the classifier confidence threshold $\tau$.

The selection score used to identify the best prototype is defined as:

$$\mathcal{S}(\widehat{\mathbf{X}}_{\text{proto}}) = P(y_{\text{target}} \mid \widehat{\mathbf{X}}_{\text{proto}}) + \text{SSIM}\left(\mathbf{X}_{\text{processed}}, \widehat{\mathbf{X}}_{\text{proto}}\right) + S_{\cos}\left(\mathbf{X}_{\text{target}}, \widehat{\mathbf{X}}_{\text{proto}}\right) \tag{3}$$

where $P(y_{\text{target}} \mid \widehat{\mathbf{X}}_{\text{proto}})$ is the target-class probability assigned by the sonar classifier, SSIM measures structural similarity between the generated prototype and the preprocessed input $\mathbf{X}_{\text{processed}}$, and $S_{\cos}$ measures the cosine similarity of pixel intensities between the generated prototype and the target sonar image $\mathbf{X}_{\text{target}}$. All three terms are bounded in $[0, 1]$. Candidates are filtered by the condition $P(y_{\text{target}} \mid \widehat{\mathbf{X}}_{\text{proto}}) \geq \tau$, and the highest-scoring prototype among them is selected. If no candidate satisfies this threshold, the globally highest-scoring candidate is returned as a fallback.

---

**Algorithm 2** Prototype-Based Explanation via Diffusion

---

**Require:** Target SONAR image $\mathbf{X}_{\text{target}}$, target label $y_{\text{target}}$, RGB prototype set $\mathcal{P}$, noise-level range $[t_{\min}, t_{\max}]$, confidence threshold $\tau$
1: **Output:** SONAR-style prototype $\widehat{\mathbf{X}}_{\text{proto}}$
2: Select an RGB prototype image $\mathbf{X}_{\text{rgb}} \in \mathcal{P}$ for class $y_{\text{target}}$
3: Remove background from $\mathbf{X}_{\text{rgb}}$ and convert to greyscale $\mathbf{X}_{\text{grey}}$
4: Apply CLAHE and add Gaussian noise $\mathcal{N}(0, 0.05)$, obtaining $\mathbf{X}_{\text{processed}}$
5: Initialise candidate pool $\mathcal{C} \leftarrow \emptyset$
6: **for** each $t \in [t_{\min}, t_{\max}]$ and augmentation $a$ (flip, scale, shift) **do**
7:     Add noise to $\mathbf{X}_{\text{processed}}$ at timestep $t$: $\mathbf{X}_t \leftarrow q(\mathbf{X}_t \mid \mathbf{X}_{t-1})$
8:     Denoise using diffusion model: $\widehat{\mathbf{X}}_{\text{proto}} \leftarrow p_\theta(\mathbf{X}_{t-1} \mid \mathbf{X}_t, y_{\text{target}})$
9:     Compute score $\mathcal{S}(\widehat{\mathbf{X}}_{\text{proto}})$ as in equation 3; add to $\mathcal{C}$
10: **end for**
11: **return** $\arg\max_{\mathcal{C}_\tau} \mathcal{S}$, where $\mathcal{C}_\tau = \{\widehat{\mathbf{X}}_{\text{proto}} \in \mathcal{C} : P(y_{\text{target}} \mid \widehat{\mathbf{X}}_{\text{proto}}) \geq \tau\}$   (fallback: $\arg\max_{\mathcal{C}} \mathcal{S}$ if $\mathcal{C}_\tau = \emptyset$)

---

## 4 Experiments & Evaluation

This section presents the experimental setup and evaluation of D-SCOPE framework for generating counterfactual and prototype explanations using diffusion models on SONAR imagery. We evaluate both the fidelity of the generated images and the closeness of the explanations to the real images, using quantitative metrics, qualitative case studies, and a user study.

### 4.1 Experimental Setup

**Datasets:**

We utilised the *Watertank* and *Turntable* datasets (Singh & Valdenegro-Toro, 2021; Valdenegro-Toro et al., 2025), both acquired under controlled SONAR conditions. From the *Watertank* dataset, we selected 8 of the original 10 classes: **bottle**, **can**, **chain**, **drink carton**, **hook**, **propeller**, **tire**, and **valve**. From the *Turntable* dataset, we selected 9 of 12 classes: **bidon**, **bottle**, **box**, **can**, **drink-carton**, **pipe**, **tire**, **valve**, and **wrench**.

We excluded some classes from both datasets due to low visual plausibility for explanations, insufficient class diversity, and/or strong background dependence. Overlapping classes across datasets (e.g., plastic-pipe) were treated as distinct due to differences in acquisition conditions. The remaining classes nonetheless include challenging cases, as shown in Figure 2, where low SNR conditions degrade object structure and class-relevant semantics. This makes training on such data challenging and increases the likelihood that low SNR samples are perceived as implausible when presented as counterfactual or prototype explanations.

All images were resized to $64 \times 64$ and normalised using standard PyTorch transforms.

**Models Configuration:** We adopted the public diffusion model codebase from (Dhariwal & Nichol, 2021). Table 1 summarises the configuration used for model training. For inference, we use timestep respacing with 160 steps; this allows the model to be significantly faster during generation. Instead of sampling at the 2000 steps used during training, we utilise a strided schedule to reduce inference time while maintaining sample quality.

Table 1: Diffusion Model Training Parameters

| Model | Image Size: 64 | Channels: 128 | Residual Blocks: 3 |
|---|---|---|---|
| **Diffusion** | Steps: 2000 | Schedule: Cosine | Class Condition: True |
| **Training** | Learning Rate: $10^{-4}$ | Batch: 128 | Weight Decay 0.05 |

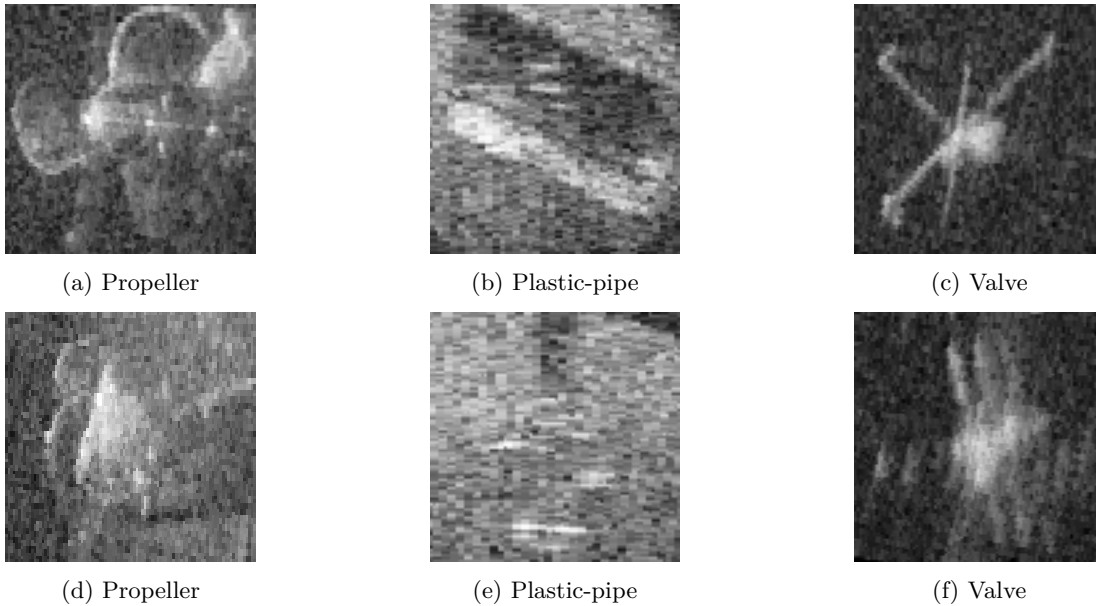

(a) Propeller  (b) Plastic-pipe  (c) Valve

(d) Propeller  (e) Plastic-pipe  (f) Valve

Figure 2: Examples from the Watertank and Turntable datasets. (Top row): High-SNR images with distinct object geometry. (Bottom row): Low-SNR images dominated by speckle noise.

Training was performed on an Nvidia RTX 3090 GPU (24GB) for approximately 24 hours. Sampling was later conducted using a lower-memory Nvidia RTX A2000 (4GB) GPU.

**Computational Latency:** D-SCOPE is an offline XAI tool, not intended for real-time use. On a consumer-grade laptop GPU (e.g., RTX A2000), generating a counterfactual takes 40s–80s depending on the query-target mapping; for example, bottle-to-can is typically faster than valve-to-propeller. Prototype generation is quicker, taking about 16s–28s.

## 4.2 Classifier Setup

To support guided diffusion and explanation generation, we trained two types of classifiers: standard classifiers on clean SONAR images and noisy classifiers on noise-corrupted images. These were tuned with Optuna (Akiba et al., 2019) and tracked with MLflow (Zaharia et al., 2018).

**Classifiers under study** We selected two architectures based on their performance and stability during training. For the *Watertank* dataset, we used a **ResNet-18**, which achieved a test accuracy of 96.49%. For the more challenging *Turntable* dataset, we utilised a **DenseNet-121**, reaching a test accuracy of 97.62%. Watertank classes exhibit greater geometric distinctiveness, while Turntable classes show realistic confusion patterns (e.g., "bidon" vs. "bottle") due to visual similarity in SONAR imagery. These results establish the baseline behaviour of the classifiers before analysing their responses to counterfactual perturbations and decision boundary transitions.

**Noisy classifier.** Following the guided diffusion framework (Dhariwal & Nichol, 2021), we trained UNet-based classifiers on noise-corrupted SONAR images ($t \in [0, 2000]$). Training on these noisy images is necessary for the classifier to provide meaningful gradients during the reverse process towards the target manifold. Since the Turntable dataset is more challenging and contains classes that are visually harder to distinguish, we increased the model capacity from a depth of 2 (128 channels) for Watertank to a depth of 4 (256 channels). Both models use attention-based pooling at resolutions of 32, 16, and 8. These noisy classifiers are used to provide class-conditional gradients that steer the reverse diffusion process toward a target class.

### 4.3 Evaluation Metrics

We evaluate the quality of generated explanations using a combination of statistical and perceptual metrics:

**Fréchet Inception Distance (FID) (Heusel et al., 2017)**: Assesses realism by comparing statistics between real and generated image distributions using features extracted from a pretrained InceptionV3 network. Lower scores indicate that the generated images are more similar in distribution to the distribution of the real images. We acknowledge that FID values may not fully reflect visual realism in sonar imagery, since InceptionV3 is pre-trained on optical imagery.

**SSIM (Wang et al., 2004)**: Assesses structural similarity at the pixel level by comparing luminance, contrast, and structure between the original and generated images. Higher SSIM values indicate more similar image structures. We focus on SSIM to evaluate counterfactual visual fidelity and do not use Learned Perceptual Image Patch Similarity (LPIPS). Although LPIPS captures fine-grained perceptual differences beyond SSIM, it relies on optical-domain features (AlexNet) that may not fully reflect human perception of SONAR images. SSIM provides a consistent and interpretable measure of structural changes, which suffices for assessing the minimal modifications needed to generate counterfactuals.

**Precision (Kynkäänniemi et al., 2019)**: Measures the fidelity of the generated images, computed using an ImageNet-pretrained `InceptionV3` network with neighbourhood size $k = 3$. It reflects the percentage of generated images that lie within the distribution of real images.

**Recall (Kynkäänniemi et al., 2019)**: Measures the diversity of the generated images. Using the same embedder setup ($k = 3$), it estimates how much of the real image distribution is covered by the generated samples.

### 4.4 Quantitative Analysis: Diffusion Models

We compare classifier-guided diffusion across different classifier scales (0.5 to 3.0). Table 2 summarises the results for 2000 generated samples, using FID, precision, and recall to evaluate image quality and diversity. As the classifier scale increases, FID scores improve, indicating more realistic image generations. However, precision and recall peak at classifier scale 1.0 and drop at higher values. This suggests that stronger guidance improves visual quality but limits diversity. Therefore, scale 1.0 provides the best balance between image quality and variety, making it a suitable setting for both counterfactual and prototype-based generation tasks.

Table 2: Quantitative Evaluation Metrics for 2000 Samples

| Scale | FID ($\downarrow$) | Precision ($\uparrow$) | Recall ($\uparrow$) |
|:---:|:---:|:---:|:---:|
| 0.5 | 3.0 | 0.37 | 0.21 |
| 1.0 | 2.8 | **0.39** | **0.31** |
| 2.0 | 2.6 | 0.37 | 0.21 |
| 3.0 | **2.5** | 0.37 | 0.21 |

### 4.5 Counterfactuals Evaluation

### 4.5.1 Quantitative analysis: Counterfactuals

We use the flip threshold (expressed as a percentage of the maximum diffusion noise) as a quantitative metric for counterfactual evaluation. It measures the minimal noise level required to flip the classifier decision from a query class to a target class. The more noise required, the harder the transformation and the longer the reverse diffusion trajectory. With timestep respacing capped at $T = 160$, we identify the minimal timestep $t^* \in [0, 160]$ at which the classifier decision flips to the target class. This timestep directly quantifies the amount of noise added during the forward diffusion and, equivalently, how much noise the reverse diffusion must remove to reach a valid counterfactual. The values reported for each query–target class mapping are averaged over 10 independent generations, resulting in 560 generations for the "Watertank" dataset and 720 for the "Turntable" dataset.

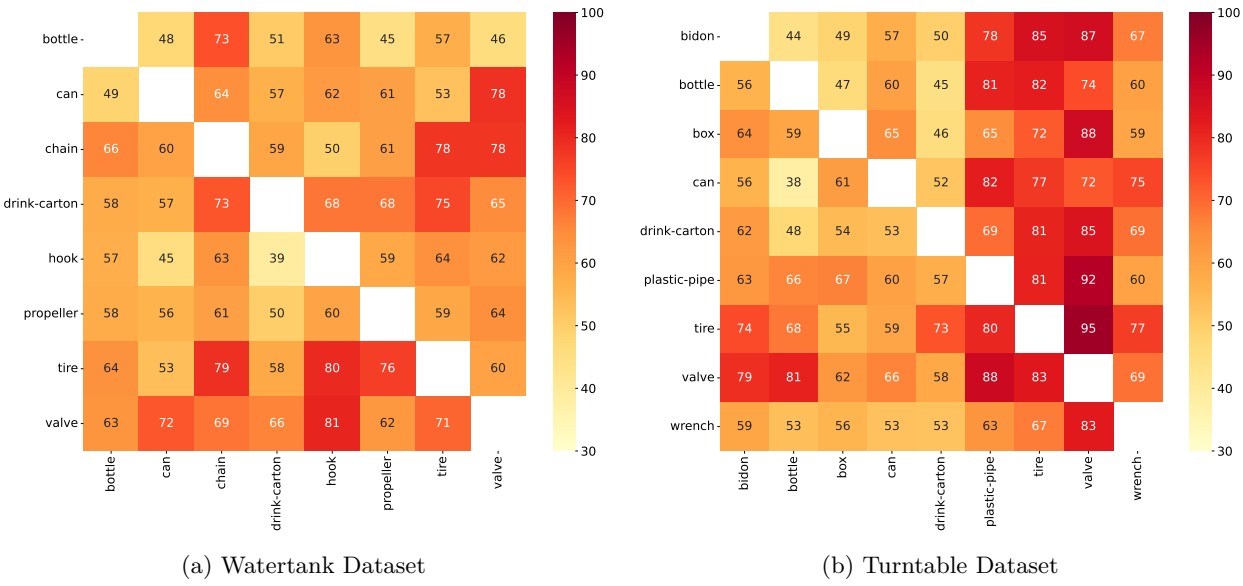

(a) Watertank Dataset

(b) Turntable Dataset

Figure 3: Query–target classifier flip thresholds. Cells show the average minimal noise level % required to flip the classifier.

Figure 3 reports this quantity as a percentage of the maximum noise level. A value of 100% corresponds to denoising from pure Gaussian noise, implying that the generated image has no structural relation to the original query image. Lower percentages indicate that the classifier decision can be flipped while preserving more of the original image structure. The "Watertank" dataset exhibits a mean flip threshold of 62% ($\sigma = 9.8\%$), while the Turntable dataset proves more challenging, with a higher mean of 65.7% ($\sigma = 13.66\%$).

The heatmaps visualise these query–target difficulties. In the "Watertank" dataset, transforming a valve query into other classes consistently requires a higher flip threshold, indicating that valves are structurally distinct and harder to modify without destroying their identity. In contrast, flipping a bottle into a valve often requires less noise. This asymmetry arises because, under certain SONAR viewpoints, valve returns exhibit elongated shapes that overlap visually with bottles, facilitating boundary crossing (see Figure 2f). Overall, bottle queries are the easiest to transform, indicating that bottle instances are more easily perturbed toward other class decision regions. For the "Turntable" dataset, the most difficult query class is the valve, while the wrench is relatively easier to transform. These results show that classifier decision boundaries can be crossed with substantially less injected noise than would be required to leave the query data manifold.

### 4.5.2 Qualitative Analysis: Counterfactuals

We analyse the transformations generated from a query image (e.g., user input) into counterfactuals targeting other classes in the two datasets. To assess structural and perceptual changes, we utilise pixel-wise difference maps and SSIM. The difference maps visualise regions of change, highlighting which areas of the image the diffusion model modified to shift the classifier's prediction from the query class to a target class.

Figure 4a presents an example using the class "**can**" as a query in the "Watertank" dataset. The leftmost column shows the original image, followed by seven counterfactuals corresponding to the remaining target classes. Below, each image is accompanied by its corresponding pixel-wise difference map, and the SSIM (per-example) value is listed under it. The lowest score is for the "**propeller**", which can be understandable since that counterfactual involves a significant structural shift, where most of the image is altered to represent a "**propeller**". On the other hand, the **drink-carton** has the highest SSIM because it looks quite similar to the original "**can**". However, perceptually, the bottle seems closer to the "can" but it's more out of focus, which results in a lower SSIM score.

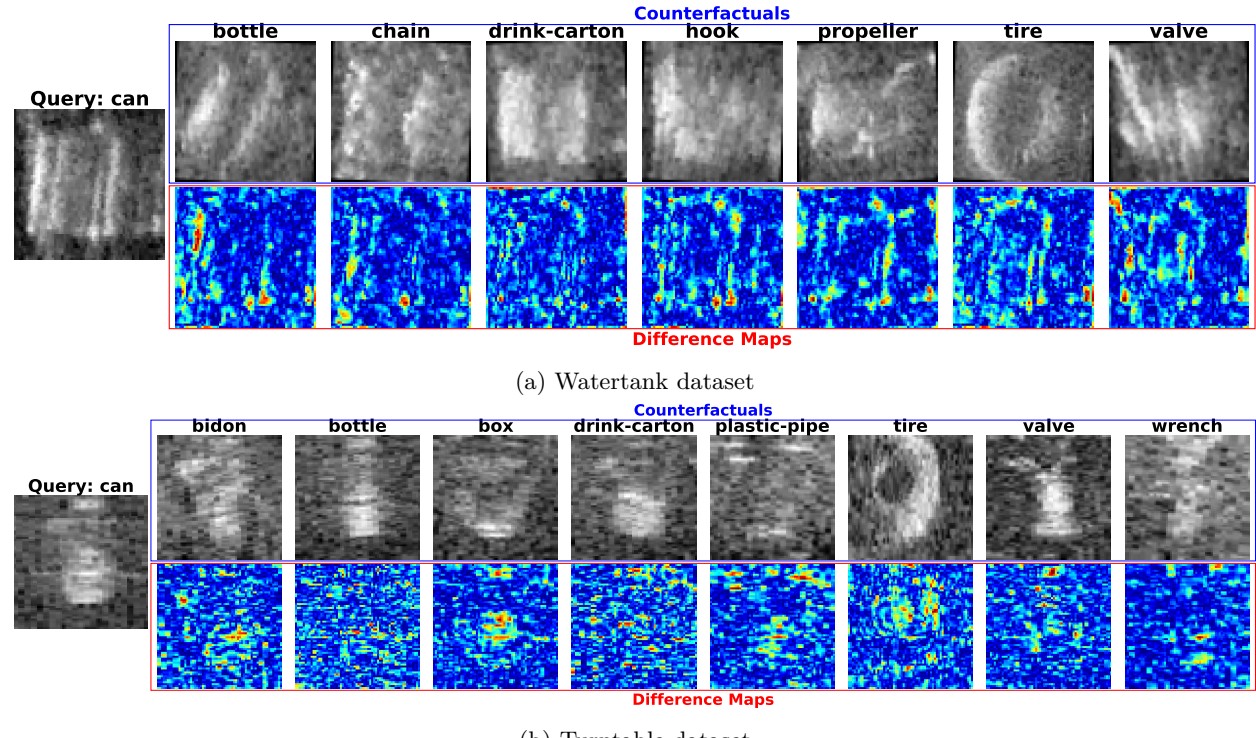

(a) Watertank dataset

(b) Turntable dataset

Figure 4: Query image of class **can** and its counterfactuals (top row) with pixel-wise difference maps (bottom row) and associated SSIM values, for two datasets.

For the "Turntable" dataset, Figure 4b shows 8 counterfactual classes against the query class "**can**". The "**bottle**" class is the nearest to the bottle with the highest SSIM compared to other classes, while the "**valve**" has the lowest similarity. For the low SNR SONAR images, D-SCOPE can successfully flip the classifier decision to a specified target-class, but the explanation could be less meaningful.

These results reflect the nature of SONAR images: they are often blurry and low in visibility, which can make it difficult to preserve structure when generating convincing counterfactuals for the user. For example, closely-related classes such as "**can**", "**drink-carton**", and **bottle** pose additional challenges. These classes often differ mainly in aspect ratios, making fine-grained changes difficult, and sometimes they can mislead the classifier. Similarly, classes such as "**chain**" and "**hook**" can be problematic, as the "**hook**" class in the dataset often includes a visible chain segment attached to the hook head. This overlap can complicate the procedure of counterfactual generation and interpretation in certain cases. These qualitative examples demonstrate that counterfactuals can provide interpretable insights for SONAR classifiers, even in cases where structural shifts are substantial.

In addition to the final counterfactual, we visualise the intermediate diffusion states preceding the prediction flip, which we refer to as SE. These states form a boundary trajectory that reveals how the input transitions from the query to the target class.

Figure 5 shows representative boundary trajectories for selected query–target pairs from the *Watertank* dataset using a ResNet-18 classifier. Each trajectory consists of successive diffusion states leading up to the first change in predicted label. When time respacing is set to 160 steps, the late stages of the trajectory become increasingly noise-dominated, resulting in a rapid loss of class-specific visual structure.

In Figures 5a and 5b, the classifier flips at timesteps where the generated samples exhibit a clear semantic alignment with the target class (e.g., bottle to valve at timestep 63). In contrast, Figure 5c illustrates a non-monotonic transition: the classifier prediction passes through intermediate classes, such as "hook", before

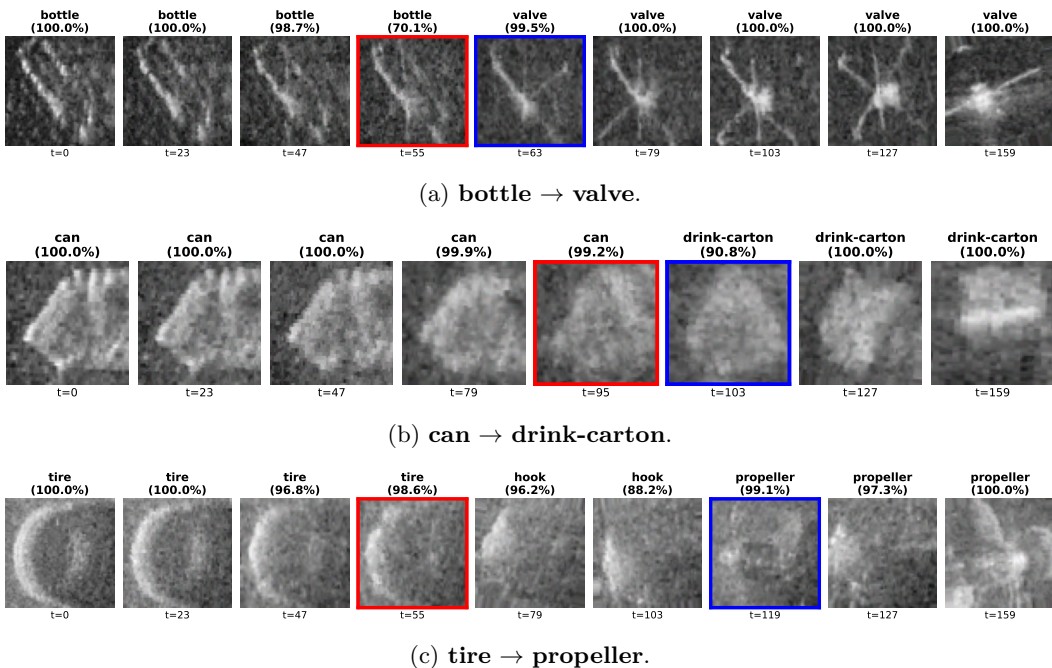

(a) **bottle → valve**.

(b) **can → drink-carton**.

(c) **tire → propeller**.

Figure 5: Boundary trajectories for selected source–target pairs in the *Watertank* dataset using ResNet-18. Semi-factual images are shown in blue and counterfactual images in red.

reaching "propeller". This occurs because certain intermediate shapes, such as crescent-like structures, are sufficient to activate the hook decision region despite limited semantic similarity to either endpoint. We nevertheless designate "propeller" as the counterfactual, as it is the first state along the trajectory that achieves the intended query-to-target flip. This example highlights that boundary trajectories need not correspond to smooth or semantically linear transformations.

### 4.6 Prototypes Evaluation

During prototype generation, we observed that increasing the input noise level in the guided diffusion process led to geometric misalignment between the input and output images. This is because the added noise covers most of the image and tends to override the input geometric structure (orientation). Because our framework lacks explicit spatial conditioning, such as conditioning the diffusion process on segmentation masks, bounding boxes, or ControlNet-style constraints (Zhang et al., 2023). We addressed this challenge by balancing the classifier guidance and the added noise to the input image to try and preserve the geometric alignment to produce semantically meaningful prototypes.

As shown in Figure 6a, the **tire** class demonstrates the most consistent prototype, as its geometry in the input RGB image closely resembles its appearance in the SONAR image. The **valve** is more challenging due to significant viewpoint differences; while the RGB image captures it from the side, the SONAR image represents a top-down perspective. To improve prototype matching for the valve, the RGB input should ideally be captured from a similar top-down view, as perceived by the SONAR. For the **hook**, our algorithm struggled when the RGB input image lacked the structural similarity to the target class in structure. This is when the RGB image only contains the hook head, while the SONAR representation includes both the hook head and an attached chain, leading to mismatched representations.

Figure 6b presents the prototype evaluation for the Turntable dataset. Similar to the Watertank setup, the **tire** class remains one of the easiest. The **valve** shows comparable behaviour across both datasets. More challenging classes include **bidon**, **bottle**, **box**, **can**, **drink carton**, and **plastic pipe**. These objects share similar SONAR appearances, with distinctions primarily arising from aspect ratio rather than shape. This

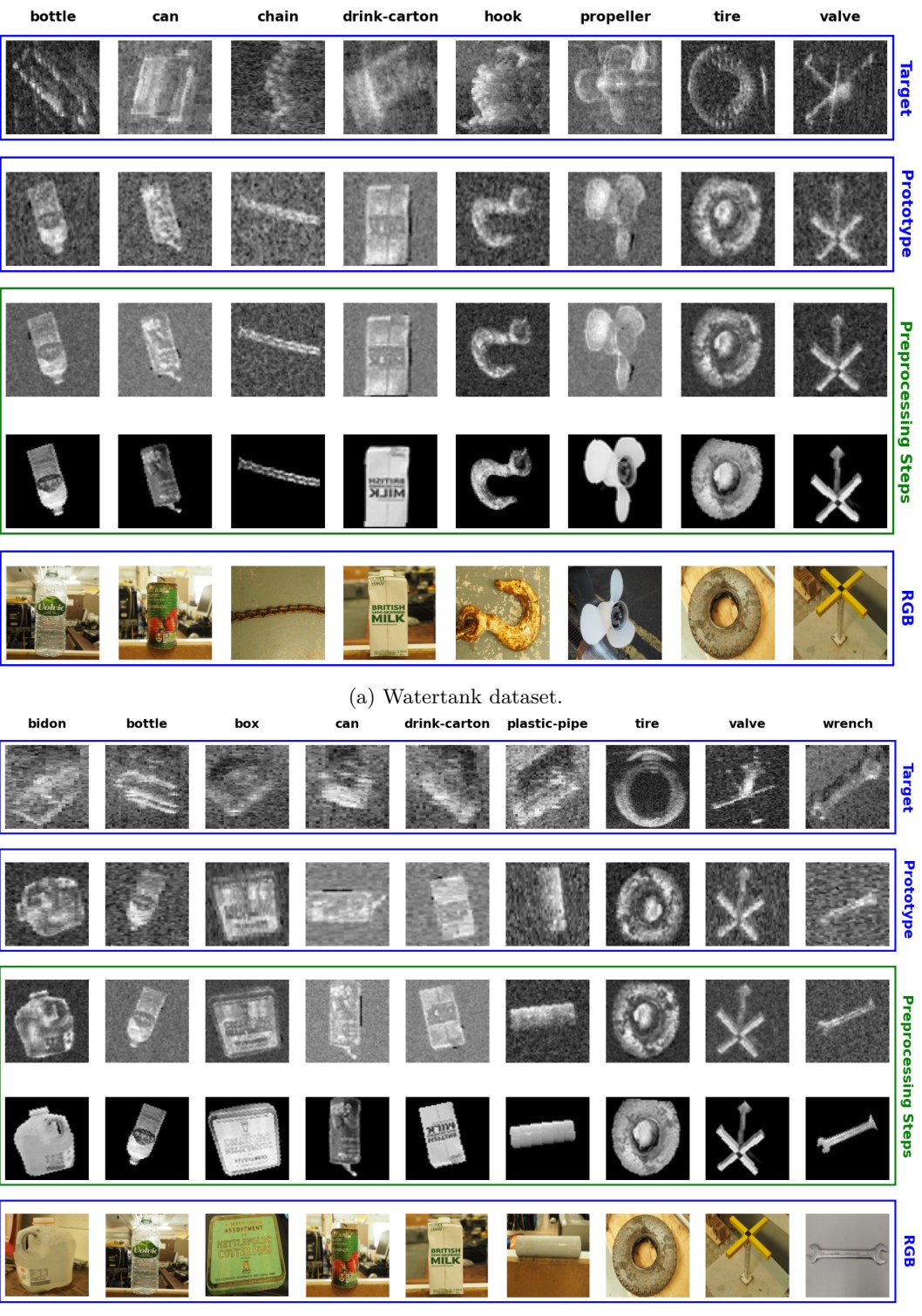

(a) Watertank dataset.

(b) Turntable dataset.

Figure 6: Prototype generation to convert from RGB to SONAR images.

suggests that PE helps bridge semantic understanding between the SONAR and RGB domains, even when object classes are visually ambiguous. The **wrench** class also demonstrates a clear and consistent prototype.

Several failure cases persist, where successful interpretation depends on the existence of a coherent and visually recognisable object structure. When the SONAR image lacks such a structure, as is common in low SNR conditions, the PE fails to establish a meaningful correspondence between the RGB and SONAR domains. This can be seen in the low SNR samples in Figure 2. Therefore, we treat these cases as dataset-driven limitations and restrict our qualitative analysis to representative cases.

### 4.7 Exploratory User Study

To gain initial insights into the perceptual interpretability of our D-SCOPE framework, we conducted an exploratory, qualitative user study with eight participants of varying machine learning experience levels, giving 128 responses in total. The participants were volunteers from a research institute. None were SONAR domain experts; therefore, the evaluation targets general perceptual interpretability rather than operational validity. Each participant assessed 16 examples sampled equally from the Watertank and Turntable datasets. The study consisted of three sequential phases. First, participants were shown the query image together with the classifier's top two predictions. They rated (i) trust in the classifier and (ii) their ability to recognise semantic structure within the query image. Then, CE or a PE were shown with randomised order. They rated the explanation according to predefined criteria. Lastly, the alternative explanation was revealed and rated in the same manner. Participants then selected which explanation better supported their understanding of classifier behaviour and provided a final trust rating.

All ratings used a 5-point Likert scale (1 = very poor, 5 = excellent). Comparative preference was recorded using six options: Strongly CE, CE, Both Equally, PE, Strongly PE, Neither. The SE were excluded to reduce cognitive load, as they are implicitly incorporated within the CE generation pipeline.

Participants assessed explanations based on Visual Realism (the plausibility and consistency of the generated image within the SONAR distribution) and Minimality (the extent to which only necessary structural changes were introduced to the query, applicable to CE only).

The presentation order of explanations was randomised to reduce priming effects and method bias.

Figure 7 summarises the results of our user study. Overall, participants rated PEs more favourably than CE. However, the difference between PE and CE ratings decreases as participants' machine learning experience increases (Figure 7a). Box plots of the four main metrics (CE Realism, CE Minimality, PE Realism, and Structure Understanding) are shown in Figure 7b. Prototype realism is much higher than counterfactual realism, reflecting the perceptual advantage of direct sequence from RGB to SONAR representation. Structure understanding varied with participant expertise and domain knowledge, suggesting that prior familiarity with ML and visual concepts affects interpretability.

We also analysed the shift in trust ratings following exposure to the explanations. As shown in Figure 7c, the D-SCOPE framework notably tended to improve trust among participants who initially reported low confidence in the classifier's predictions. PEs were generally easier for end-users to interpret, as they operate in the RGB domain and visually illustrate the transition to the SONAR domain, whereas CEs typically require more specialised expertise and domain knowledge. These results indicate supporting evidence that perceptually realistic explanations can enhance user trust and interpretability, particularly for participants with limited prior experience.

### 4.8 Qualitative Reference: Attribution vs. Generative Explanations

To contextualise D-SCOPE within the broader interpretability landscape and illustrate how different explanations behave on SONAR data, we visualise three established methods: GradientSHAP Lundberg & Lee (2017), GradCAM Selvaraju et al. (2017), and LIME Ribeiro et al. (2016), implemented using the Captum library Kokhlikyan et al. (2020). Figure 8 displays their outputs alongside D-SCOPE explanations.

Each attribution method captures a different aspect of how the model responds to its input. GradientSHAP estimates pixel importance by comparing gradients against random baselines, assigning each pixel a positive or negative contribution to the predicted class. GradCAM highlights the most influential image regions by upsampling gradient-weighted activations from the final convolutional layer, producing a coarse heatmap

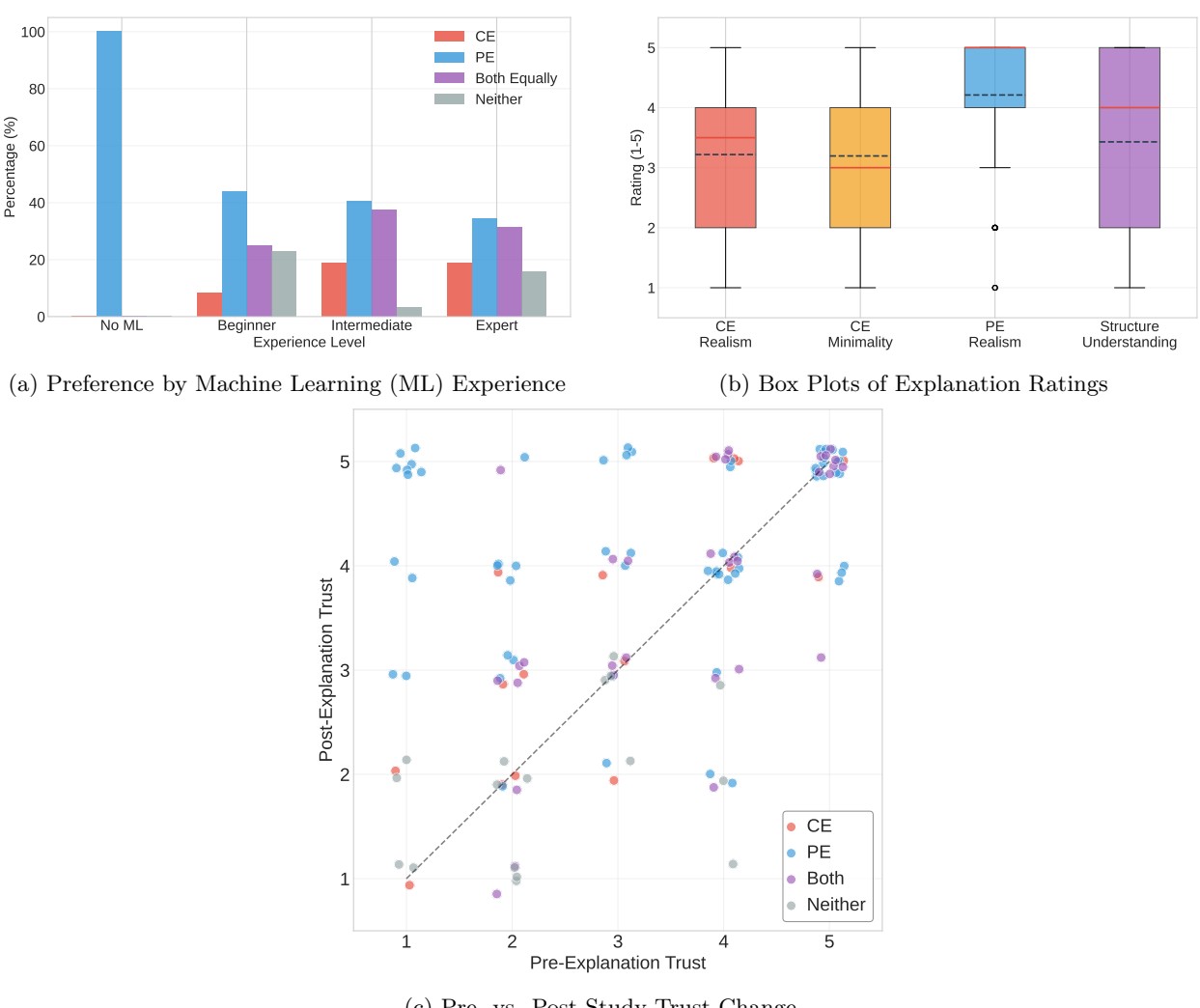

(a) Preference by Machine Learning (ML) Experience

(b) Box Plots of Explanation Ratings

(c) Pre- vs. Post-Study Trust Change

Figure 7: Summary of human evaluation results.

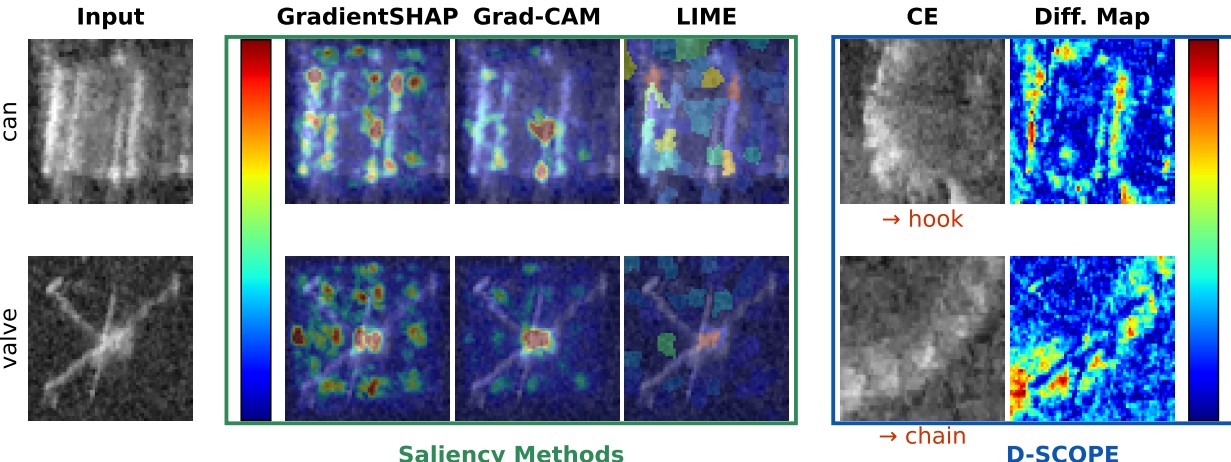

Figure 8: **Comparison of explanation types on SONAR data.** Saliency-based baselines (GradientSHAP, GradCAM, LIME) highlight local pixel importance (blue = low, red = high attribution). D-SCOPE generates CE paired with a pixel-change difference map (blue = low, red = high change), providing a contrastive, class-targeted explanation grounded in the data manifold.

of discriminative areas. LIME explains predictions by perturbing superpixel regions and fitting a simple interpretable model to approximate the classifier's local decision boundary.

These approaches simply answer different questions. Attribution methods show the model's focus by highlighting influential input regions. D-SCOPE offers a more natural way for end-users to interpret these outputs by operating at a complementary, structural level. By producing counterfactual images and cross-domain prototypes directly in the acoustic domain, the framework provides class-conditioned visual references that characterise the decision boundary itself rather than a single prediction. Figure 8 illustrates this distinction: the attribution baselines yield abstract importance masks, while D-SCOPE generates intuitive image modifications that expose the structural differences the model uses to separate classes.

## 5 Conclusions

This work introduced D-SCOPE, a post-hoc explainability framework that provides counterfactual (CE), semi-factual (SE), and prototype-based (PE) explanations for SONAR imagery. D-SCOPE does not require access to the internal structure or parameters of the classifier. Instead, it relies on class-conditional diffusion models to generate visual explanations. We evaluated it on two marine debris datasets: *Turntable* and *Watertank* (Valdenegro-Toro et al., 2025).

This approach provides the end user with a practical solution for explainability by addressing "what-if" and "even if" scenarios. CE shows how a query image would need to change to be classified as a target class, while SE captures intermediate states before the prediction flip to reveal how gradual perturbations influence the classifier's decision. Additionally, PE maps familiar optical-domain examples into the SONAR modality, allowing users to reason about predictions through cross-domain visual analogies. A user study with non-experts in the SONAR domain further confirmed that these explanations, particularly prototypes, enhance interpretability and can improve trust in the classifier's predictions.

Several challenges remain for future exploration. Extending this framework to unknown classes is difficult, as both the diffusion model and the classifiers currently require access to the training data. Future work could explore generalisation to unseen data, possibly by leveraging zero-shot approaches. Additionally, while the D-SCOPE framework is validated on underwater SONAR data, its underlying generative design is well-suited for other non-optical active sensing modalities that suffer from low SNR and speckle noise. Future work will investigate the transferability of this framework to Synthetic Aperture Radar (SAR) imagery. Furthermore, the visual similarity metrics used in D-SCOPE have additional limitations. SSIM is highly sensitive to

alignment; minor flips, rotations, or translations can significantly distort the metric. Additionally, the FID backbone network needs fine-tuning specifically for the SONAR modality for more precise evaluation.

We refer to our approach (D-SCOPE) as visual explainability, as it provides explanations by giving examples or counterfactuals of the classifier's output decision. However, not all SONAR data can be effectively interpreted through visual means alone. In several cases, particularly in the turntable dataset, certain SONAR images appear as near-random noise to the human eye, despite containing meaningful acoustic patterns. This highlights the potential need for acoustic explainability, where explanations are derived from signal-level properties rather than visual appearance. Future work could explore this direction by incorporating domain-specific acoustic attributes, such as object-specific backscatter characteristics or estimated acoustic impedance, to explain how sound interacts with underwater materials and shapes. Such an approach could complement visual explanations and provide interpretability even when visual cues are absent or ambiguous.

## 6 Broader Impact

The primary positive impact of D-SCOPE is environmental. By making SONAR classifications interpretable to end-users through counterfactuals and cross-domain prototypes, this framework improves the efficiency and reliability of civilian marine debris monitoring and ocean cleanup operations. There is also high potential to extend this framework to other civilian applications beyond debris detection, such as Search and Rescue (SAR) operations or assisting autonomous underwater robots tasked with ensuring diver safety.

However, SONAR is a dual-use technology. While this framework is trained and validated exclusively on public civilian marine debris datasets, the underlying diffusion-based explainability mechanisms could theoretically be adapted to military applications, such as automated naval target recognition or mine countermeasures. To mitigate the risk of proprietary military exploitation, we release our code and models openly to ensure these interpretability advancements remain accessible to the civilian scientific community for ecological and humanitarian research.

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
