# OpenReview forum: "D-SCOPE: Diffusion-based Sonar Counterfactual and Prototype Explanations"
_TMLR — Decision pending for TMLR_

### Review · Reviewer_o2fR · 2026-04-12

**Summary Of Contributions:**

Diffusion models have recently expanded the toolkit for explainable and interpretable machine learning in computer vision. In this work, the authors adapt guided diffusion models to SOund NAvigation and Ranging (SONAR) imagery and propose D-SCOPE, a post-hoc explainability framework for SONAR image classification. The framework contains two main components. First, a counterfactual / semi-factual explanation module generates modified versions of a query image conditioned on a user-specified target class, allowing users to explore “what-if” scenarios and inspect the intermediate changes that lead to a prediction shift. Second, a prototype-based explanation module constructs SONAR-style class prototypes from predefined reference examples, providing more intuitive visual exemplars for understanding classifier outputs. Experiments suggest that the generated images are visually realistic and that the proposed framework can improve the interpretability of SONAR classification results.

**Audience:**

No

**Audience Explanation:**

I am uncertain whether this paper will strongly interest the typical TMLR audience. My main concerns are its novelty and its narrow domain specificity.

On the novelty side, while the paper is among the earlier works to study interpretability for the SONAR modality, the modality itself does not appear to differ from standard computer vision settings in a way that motivates substantial methodological innovation beyond increased noise, lower resolution, and grayscale imagery. Many of such images have been well studied in the earlier times of computer vision. The two main components of the framework largely adapt existing diffusion-based explanation ideas to this new domain, with limited SONAR-specific methodological development. As a result, the paper’s originality feels relatively modest, and I am not convinced that it offers enough new insight for a broad machine learning audience.

The paper’s specificity to SONAR is also a concern. The work may be more compelling to researchers in marine engineering or underwater sensing than to the general TMLR readership. For instance, most of section 4.5 focuses more on observations about the dataset and SONAR class relationships than on deeper analysis of the proposed explainability method itself. While these findings may be useful, they seem more valuable to domain specialists working directly with SONAR imagery than to machine learning practitioners looking for broadly transferable methodological advances. For this reason, I suspect the paper may be better suited to a venue closer to the marine sensing or applied engineering community.

Other concerns regarding this paper:

Several figures could be improved substantially. In particular, Figures 1 and 4 are difficult to read because the text is too small, to the point that I had to zoom in significantly just to inspect them properly. Improving figure readability would make the paper much easier to follow.

**Broader Impact Concerns:**

A Broader Impact statement would be needed for this paper. While the primary focus of the paper is on civilian marine debris finding, the SONAR technology and images is dual-use. Methods for interpreting SONAR-based object classification could be relevant to military applications, and it will have impact on the society.

**Claims And Evidence:**

Yes

**Claims Explanation:**

The authors conduct quantitative experiments in Section 4.4 to evaluate the fidelity and quality of the generated images, and they further complement these results with a human evaluation in Section 4.7 to assess the interpretability of the proposed explanations.

**Requested Changes:**

The paper would require **substantial revision** before it would be a strong fit for TMLR. As stated above, in its current form, the work is framed primarily around the SONAR application domain and appears more relevant to marine engineering or underwater sensing researchers than to a broad machine learning audience. To improve its suitability for TMLR, the authors would need to better **introduce and highlight the methodological contributions** for machine learning practitioners, clarify what is genuinely novel beyond applying existing diffusion-based explanation methods to a new modality, and provide stronger evidence that the proposed ideas offer transferable insights beyond this specific application setting.

---

> ### Author Response · Authors · 2026-05-21
> **Official Comment by Authors**
>
> We thank the reviewer for their careful reading and constructive feedback. We have revised the manuscript accordingly and addressed each concern below.
>
>
> ### Main Concern: Novelty, Domain Generality, and Fit for TMLR
> We understand the concern that the paper could be read as a domain-specific application of existing diffusion ideas. While the manuscript is centred on SONAR, we use it as a testbed precisely because it is a highly challenging, low-SNR, low-resolution sensor modality. We have updated the introduction to clarify this broader methodological relevance to low-SNR data.
>
> The underlying explanation framework is modality-agnostic and can be extended to other low-SNR sensor imagery, including SAR (Synthetic Aperture Radar). To explicitly address domain generality, we have added a paragraph to the discussion section outlining how the method maps to SAR data characteristics.
>
> ### Figures and Readability
>
> Thanks for pointing out the font size of Figures 1 and 4. We have updated both figures in the revised manuscript to increase the text size and improve readability.
>
> ### Broader Impact
>
> We agree that a Broader Impact statement is needed. The paper’s primary use case is civilian marine-debris detection, but SONAR-based classification and its interpretation can also be relevant in dual-use contexts. The statement explicitly discusses this dual-use risk.

---

### Review · Reviewer_1bBu · 2026-04-13

**Summary Of Contributions:**

This work considers the interpretability of classifiers for sonar images. It proposes the use of diffusion models for this purpose. The framework generates two types of explanations: counterfactual-based and prototype based. The former involves running the diffusion process and identifying the smallest step at which the class flips -- providing this modified image as an explanation of the features that would cause the classifier output to change. The latter involves transforming the image to a prototypical example for each class. The method is evaluated quantitatively and qualitatively, including a small-scale evaluation with human participants.

**Additional Comments:**

Additional comments and nits:

- Why is SSIM not assessed in Table 2 despite being introduced as a metric in Section 4.3? The way the structure is set up makes the reader expect it.
- The writing style of the paper tends to be overly lengthy. The abstract, for example, is much too long -- as are the paragraphs in the introduction.
- Mixed consistency of SONAR vs "sonar".
- Algorithm 1: newline before return in line 19?

**Audience:**

No

**Audience Explanation:**

A1. **Narrow scope**: the scope of the paper is very narrow -- in essence, generating explanations for a specific application (marine debris classification).  There will be limited interest from researchers not working with sonar data more broadly.

A2. **Lack of methodological contributions**: if my understanding is correct, the paper applies existing interpretability techniques to this domain, and does not provide a meaningful methodological contribution.

**Claims And Evidence:**

No

**Claims Explanation:**

C1. **Lack of baselines**: the proposed methods are not compared to any other interpretability technique. A comparison should be presented with more standard techniques such as saliency maps, especially given the complexity of setting up and running the diffusion-based methods. The fact that existing techniques are limited, as claimed, should be validated with evidence.

C2. **Small N**: the study with human participants is very small-scale (N=8). No statistically robust evidence is provided.

C3. **Magic numbers**: the methods are very complex and requires setting various parameters (e.g., the diffusion parameters in Algorithm 2, the weights in equation 3, among many others). It is unclear how robust the method are to these choices since no sensitivity analysis is performed. It is also unclear how well the method would transfer to another application using sonar data beyond classifying marine debris, since the two datasets tested on are highly related.

C4. **Soundness of prototype generation**: the fact that sonar data is fundamentally different from image data is part of the premise of the paper. So why are the chosen prototypes derived by transforming image data instead of selecting prototypes that are already in the sonar modality? While visually appealing, these explanations are potentially unsound in my opinion.

**Requested Changes:**

All of C1-C4 should be addressed in order to secure acceptance, though this is not feasible for a rebuttal cycle in my opinion. Additionally, concerns A1 and A2 still stand given the typical quality bar of TMLR.

---

> ### Author Response · Authors · 2026-05-21
> **Official Comment by Authors**
>
> Thank you for your very detailed review of our paper. We sincerely appreciate your feedback.
>
> ---
>
> ### C1. Lack of baselines:
>
> We have added a baseline comparison that runs GradientSHAP, GradCAM, and LIME on the same classifier and input images used for the counterfactual explanations. The results are collected in a new figure in the revision. These attribution methods and the counterfactual explanations answer different questions: saliency and GradCAM indicate which pixels drive the current prediction, while counterfactuals show what the image would need to look like for the decision to change. We present both side-by-side precisely to make this distinction concrete, rather than to claim one is better than the other.
>
> ### C2. Small N:
>
> We agree that $N=8$ is small, and we do not claim statistical significance. To align with feedback from another reviewer, we have completely reframed this section as an exploratory, qualitative user study and updated the subsection title to 'Exploratory User Study' to make it clear this is a preliminary, qualitative analysis.

---

> > ### Author Response · Authors · 2026-05-21
> > **Official Comment by Authors**
> >
> > ### C3 and C4. Magic numbers and soundness of prototype generation
> > Thanks for pointing these out. Both concerns are addressed by a revised prototype methodology. We updated Algorithm 2 and
> > replaced Equation 3 with a parameter-free selection score, so that the chosen prototype is directly determined by classifier confidence and sonar-domain fidelity without empirically tuned weights.

---

> > > ### Author Response · Authors · 2026-05-22
> > > **Official Comment by Authors**
> > >
> > > ### A1 Narrow Scope:
> > > The reviewer's point regarding this is well taken. The manuscript has been revised in the Introduction and Conclusion to clarify that the proposed architecture is intended generally for low-Signal-to-Noise Ratio (SNR). The SONAR domain is utilised as a highly constrained testbed to evaluate the framework rather than defining its ultimate boundary. The text now notes potential transferability to structurally similar data types, such as Synthetic Aperture Radar (SAR), while strictly limiting the claims to what has been empirically verified.
> > >
> > >
> > > ### A2. Methodological contributions
> > >
> > > We acknowledge the reviewer’s perspective regarding the application of existing generative techniques. However, we would like to clarify the specific methodological integration presented in this work. D-SCOPE not only apply standard interpretability tools to a new dataset; rather, it introduces a unified framework that structurally combines counterfactuals, semi-factual trajectories, and cross-domain prototype-based explanations within a single pipeline applied to low SNR imagery.

---

> > > > ### Author Response · Authors · 2026-05-22
> > > > **Official Comment by Authors**
> > > >
> > > > ### Additional Comments:
> > > > We thank the reviewer for these helpful observations. We have updated the manuscript as follows:
> > > >
> > > > * Regarding SSIM in Table 2: Table 2 is used strictly for hyperparameter tuning (selecting the classifier scale) using standard generative metrics. SSIM is reserved for evaluating the final generated explanations later in the paper.
> > > >
> > > > * Regarding Consistent Casing: We have standardised the casing to "SONAR" throughout the manuscript.
> > > >
> > > > * Regarding Algorithm 1: We fixed the formatting in Algorithm 1 by adding a newline before the return statement in line 19.

---

### Review · Reviewer_ns4H · 2026-05-07

**Summary Of Contributions:**

The authors have trained a class conditional diffusion model based on a open dataset of pixel space image reconstructions of sonar data. Using this diffusion model, they have come up with 2 methods to generate visual explanations of image classifiers for this dataset (1) counterfactual explanations and (2) prototypical explanations. They describa nd analyze these methods.

The first method is very similar in spirit to https://arxiv.org/pdf/2411.15265 that they refer to in related works, but is somewhat simpler to follow. One takes an image X classified into class C and corrupts with noise. Through guided diffusion, the image is denoised towards class C’. By varying the degree of noise added, the denoised image can be more like X or more like some X’ maximally corrupted and denoised. The end user is presented with images in the range from X to X’, annotated with classification results. This provides the user with qualitative insights in what image properties flips the classification.

The second method seems novel. Given a class C they generate samples X_C from that class. To this end, they take RGB photographies of objects in class C, preprocess them to look more like sonar images, add noise, and denoise towards class C. The classifier and the original image X takes no part in this process. One can argue if this reallt sheds inisghts into the classifier model, or if this is a separate effort from explainability of the classifier.

I find the methods quite inventive, and I can see how they are useful in similar interpretation and explainability work for image classifiers. I don’t see that the proposed methods will have much impact on machine learning practices, I don’t think it is a very good fit for TMLR. A venue more specialized in applied image analysis or applied explainable AI could be more suitable.

**Audience:**

No

**Audience Explanation:**

I think that the presented results has too low generalizibility to be interesting enough. If the survey with human subjects were repeated with sonar experts, I think this would be a great improvement and be of more interest. If the methods were framed in more generality, or if more parts of the system could be motivated by regirous mathematical analysis, that could also raise the interest in the work.

The human evaluation at the end holds some merit, in that it investigates human preferences in XAI methods. This could, if expanded, have impact on directions for what XAI methods to pursue, but I find the current version of the investigation too small scale and unstructured.

**Claims And Evidence:**

Yes

**Claims Explanation:**

The claims are quite limited. The authors claim to propose two methods to generate visual explanations, and qualitative, quantitative, and ‘human’ evaluations thereof.  Indeed this is what they do.

The methods are described in algorithm displays with sufficient clarity to follow. Some parts should be cleared up, such as selecting the ‘best’ X_{cf} in line 18 of Algorithm 1, not explicitly giving the criterion for what ‘best’ means. Last paragraph page 6 indicates that classes might not only be the object type (e.g. ‘propeller’ and ‘valve’) but also other qualitative properties, such as material. This is unclear writing and confusing since this is not referenced at any other place in the paper, but does not really change any parts of the proposed method. In equation 3 symbols are used that are first introduced on the next page, in Algorithm 2. Several others small notational problems like this are present throughout.

There are also unclear parts of the evaluation. For the Precision and Recall metrics, an embedder is needed. It is not explained what embedder is used. Nor what k is in the kNN step of the algorithm in the cited Kynkäänniemi et al., 2019. These are  not very severe errors, but must be adressed.

The evaluation by humans is analyzed and discussed with appropriate care, and I think the claims about it are correct. But the study population is not described well, and their affiliation with the research is unclear. Considering the discrepancy in answers between non-ML-trained users and others, I wonder about the extrernal validity of the findings. This part is the methodologically and scientifically weakest part of the manuscript.

**Requested Changes:**

for me to support acceptence, you must raise the generalizability of the findings. I think you must do at least one of:
- mathematical analysis of the properties in the counterfactual analysis. what can be said about the path in image space of reconstructed images from X_{target} to X_{cf}?
- link the prototypical explanations to properties of the classifier itself. the data manifold learned of the diffusion model -- how does it relate to the classification boundaries of the classifier? how do we know that the prototypes are prototypical classification examples, and not just prototypical examples in the data manifold of the diffusion model?
- expand the user study. survey actual end users. make sure they are unbiased in their assessments. do either a larger scale statistical analysis, with appropriate power to detect any relevant differences in preferences. or change methodology to a focus group / qualitative method observational user study.

after that, one could discuss smaller issues like typography.

---

> ### Author Response · Authors · 2026-05-21
> **Official Comment by Authors**
>
> We sincerely thank the reviewer for the constructive and detailed feedback. We have carefully addressed the comments, and below is a summary of our revisions:
>
>
> ### Venue fit
> We understand the concern. The sonar domain is undeniably applied, but we believe the methodological contributions, the semi-factual boundary sweep and the empirical link between classifier-guided prototype generation and classifier decision regions are framed at a level of generality that extends to any class-conditional diffusion model used for explanation. We have made this framing more explicit in the revision. We also added a paragraph about the SONAR as being a test bed for low-SNR sensors, and that potentially D-SCOPE can be extended to other low-SNR sensors such as SAR (Synthetic Aperture Radar).
>
> ### Material / shape annotation paragraph (page 6):
> We thank the reviewer for flagging this. The paragraph was unintentionally left in from an earlier version of the manuscript, where ontology-based annotation was planned. This feature is not implemented in the current system, and the paragraph has been removed in the revised manuscript. We apologise for the confusion.
>
> ### Precision, Recall, and embedder:
> We agree that these were not mentioned explicitly. The revision now states that both metrics use an ImageNet-pretrained InceptionV3 network as the embedder with neighbourhood size k=3, following Kynkäänniemi et al. (2019).

---

> > ### Author Response · Authors · 2026-05-22
> > **Official Comment by Authors**
> >
> > Thanks for the constructive feedback:
> > ### Requested Changes:
> >
> > * We have added a formal mathematical characterization of the counterfactual path in Section 3.2 of the revised manuscript. We formalise the sequence as a discrete trajectory and demonstrate that it represents a non-linear, manifold-constrained path that minimizes structural displacement while crossing the target decision boundary.
> >
> > * We have revised the prototype methodology. We updated Algorithm 2 and replaced Equation 3 with a parameter-free selection score, so that the chosen prototype is directly determined by classifier confidence and sonar-domain fidelity without empirically tuned weights.
> >
> > * We took your suggestion into consideration, and we have reframed this section as an exploratory, qualitative user study and updated the subsection title to 'Exploratory User Study' to make it clear this is a preliminary, qualitative analysis